# ChAdOx1 nCoV-19 (AZD1222) or nCoV-19-Beta (AZD2816) protect Syrian hamsters against Beta Delta and Omicron variants

Neeltje van Doremalen [1] ✉, Jonathan E. Schulz[1], Danielle R. Adney [1,8], Taylor A. Saturday[1], Robert J. Fischer [1], Claude Kwe Yinda[1], Nazia Thakur [2,3], Joseph Newman [2], Marta Ulaszewska[3], Sandra Belij-Rammerstorfer[3], Greg Saturday [4], Alexandra J. Spencer [3], Dalan Bailey [2], Colin A. Russell [5], Sarah C. Gilbert [3], Teresa Lambe[6,7] & Vincent J. Munster [1] ✉

ChAdOx1 nCoV-19 (AZD1222) is a replication-deficient simian adenovirus–vectored vaccine encoding the spike (S) protein of SARS-CoV-2, based on the first published full-length sequence (Wuhan-1). AZD1222 has been shown to have 74% vaccine efficacy against symptomatic disease in clinical trials. However, variants of concern (VoCs) have been detected, with substitutions that are associated with a reduction in virus neutralizing antibody titer. Updating vaccines to include S proteins of VoCs may be beneficial, even though current real-world data is suggesting good efficacy following boosting with vaccines encoding the ancestral S protein. Using the Syrian hamster model, we evaluate the effect of a single dose of AZD2816, encoding the S protein of the Beta VoC, and efficacy of AZD1222/AZD2816 as a heterologous primary series against challenge with the Beta or Delta variant. Minimal to no viral sgRNA could be detected in lungs of vaccinated animals obtained at 3- or 5- days post inoculation, in contrast to lungs of control animals. In Omicron-challenged hamsters, a single dose of AZD2816 or AZD1222 reduced virus shedding. Thus, these vaccination regimens are protective against the Beta, Delta, and Omicron VoCs in the hamster model.

At the end of 2019, the causative agent of COVID-19, severe acute respiratory syndrome coronavirus 2 (SARS-CoV-2), was first detected in Wuhan, China[1,2]. As of February 2nd 2022, SARS-CoV-2 has infected an estimated 380 million people, causing more than 5 million deaths[3]. Its emergence prompted the rapid development of vaccines based on the viral receptor binding protein, spike (S)[4–6]. Several vaccines demonstrated efficacy through clinical trials in less than a year[7–11] and were approved for emergency use by different regulatory bodies worldwide. Over 4.4 billion people are estimated to have received at least one dose of COVID-19 vaccination[3]. One of those vaccines is AZD1222 (ChAdOx1 nCoV-19), developed by Oxford University and produced by AstraZeneca. AZD1222 is a replication-deficient simian

[1]Laboratory of Virology, National Institute of Allergy and Infectious Diseases, National Institutes of Health, Hamilton, MT, USA. [2]Viral Glycoproteins Group, The Pirbright Institute, Pirbright, Woking, UK. [3]Pandemic Sciences Institute, Nuffield Department of Medicine, University of Oxford, Oxford, UK. [4]Rocky Mountain Veterinary Branch, National Institute of Allergy and Infectious Diseases, National Institutes of Health, Hamilton, MT, USA. [5]Laboratory of Applied Evolutionary Biology, Department of Medical Microbiology, Academic Medical Center, University of Amsterdam, Amsterdam, The Netherlands. [6]Oxford Vaccine Group, Department of Paediatrics, University of Oxford, Oxford, UK. [7]Chinese Academy of Medical Science (CAMS) Oxford Institute (COI), University of Oxford, Oxford, UK. [8]Present address: Lovelace Biomedical Research Institute, Department of Comparative Medicine, Albuquerque, NM, USA. ✉e-mail: neeltje.vandoremalen@nih.gov; vincent.munster@nih.gov

adenovirus–vectored vaccine encoding the non-stabilized S protein of Wuhan-1, one of the first published full-length SARS-CoV-2 sequences[12]. AZD1222 was shown to be highly effective in clinical trials, demonstrating 74% vaccine efficacy against symptomatic disease[7]. A two dose primary series of AZD1222 is approved for usage in more than 170 countries, and more than two billion doses of vaccine have been distributed worldwide[13].

Despite the development and administration of these vaccines, a large portion of the world's population is still unvaccinated, particularly in low-income countries[3]. Furthermore, although COVID-19 vaccination is protective against severe disease, it is not fully protective against infection with SARS-CoV-2, and breakthrough infections regularly occur[14]. High levels of circulating virus, asymptomatic infections, low vaccine coverage and break-through infection together means SARS-CoV-2 continues to circulate in the population. As a consequence, several variants of concern (VoCs) have been detected. A variant is termed a VoC if it is associated with an increase in transmission or virulence, or a decrease in the effectiveness of public health and social measures, such as diagnostics, vaccines, or therapeutics[15]. Since COVID-19 vaccines were developed early in the pandemic, they are based on the ancestral S protein, and substitutions in S may result in a reduced vaccine efficacy against VoCs. Several VoCs have substitutions in the receptor binding domain of S which are associated with a reduction in neutralizing virus titers[16–20], which is a strong predictor of vaccine efficacy[21], but the current vaccines clinically available are largely able to protect against severe disease and hospitalization caused by VoCs[22,23]. Here, we investigate the efficacy of an updated vaccine based on the S protein of the Beta variant (AZD2816)[24] against three different VoCs; the Beta, Delta, and Omicron variants, relative to the original AZD1222 vaccine.

## Results

### Vaccination of hamsters with AZD2816 result in robust humoral response

We vaccinated 44 Syrian hamsters (18 M, 26 F) either with a single dose of AZD2816 (prime only group), or with a prime dose of AZD1222 followed by a boost dose of AZD2816 (prime-boost group), or with two injections of ChAdOx1 GFP (control group) (Fig. 1A). On day 0, eight (F) hamsters per group were euthanized and serum was collected. Binding antibody titers against S protein were determined. In the prime-boost group, high binding titers were detected against all three S proteins. In the prime only group, binding antibody titers were significantly higher against Beta S compared to ancestral S (Fig. 1B).

Virus neutralizing (VN) antibody titers were determined using both a lentivirus-based pseudotype as well as live (whole SARS-CoV-2) VN assay. Pseudotype VN titers were significantly lower against Omicron in both vaccine groups. In the prime only group, pseudotype VN titers were significantly lower for Delta than ancestral S (Fig. 1C). Live VN titers were significantly higher against the Beta VoC compared to the Delta VoC in both vaccinated groups (Fig. 1D).

Using the pseudotype VN assay, the influence of single substitutions in the S protein compared to ancestral S protein were investigated: K417N, N501Y (present in the Beta and Omicron VoC), E484K (present in the Beta VoC) and L452R (present in the Delta VoC). Significantly higher VN titers were found against the E484K mutant compared to ancestral S in prime only sera, but no other differences were found (Supplementary Fig. 1A).

Together, these data show that a single vaccination with AZD2816 induces a robust immune response against S protein in hamsters, specifically against the Beta S or an S with the E484K mutation. However, vaccination with AZD1222 followed by AZD2816 does not result in

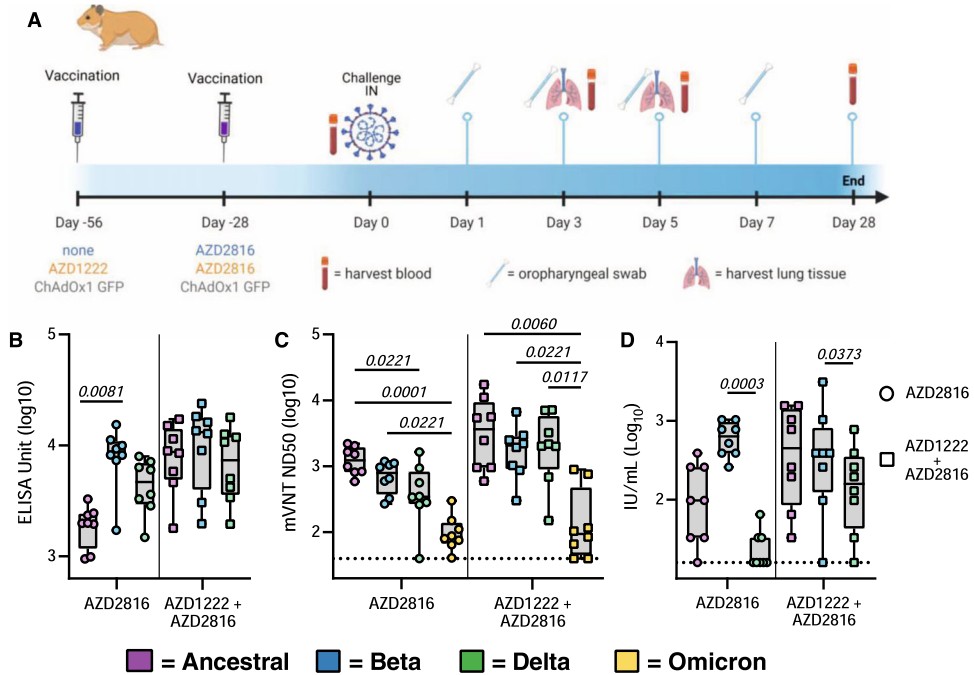

**Fig. 1 | Humoral response of Syrian hamsters vaccinated with a single dose of AZD2816, a prime dose of AZD1222 and boost dose of AZD2816, or a prime-boost regimen of ChAdOx1 GFP. A** Schematic overview of experiment. Hamsters were vaccinated with AZD2816 on day −28, AZD1222 on day −56 and AZD2816 on day −28, or ChAdOx1 GFP at day −56 and −28. Created with BioRender.com. **B** Boxplot of binding antibodies against S protein of SARS-CoV-2 (ancestral, Beta, or Delta) in serum obtained on day 0. **C** Boxplot (minimum to maximum) of pseudovirus neutralizing antibody titers in hamster sera obtained on day 0 against ancestral, Beta, Delta, or Omicron VoC. Dotted line = limit of detection. **D** Boxplot (minimum to maximum) of virus neutralizing antibody titers in hamster sera obtained on day 0 against ancestral, Beta, or Delta VoC. VN titers were normalized against NIBSC standard. Dotted line = limit of detection. Statistical significance was determined via a Friedman test followed by Dunn's multiple comparisons test, p-values in italic when significant. N = 8 per group, circles = hamsters vaccinated with AZD2816, squares = hamsters vaccinated with AZD1222 followed by AZD2816. All boxplots are drawn from first quartile to third quartile, with a line at the median. Whiskers go from each quartile to minimum or maximum values. Source data are provided as a Source Data file.

a significant increase in binding to or neutralization of Beta or E484K S compared to ancestral S.

To better understand the antigenic distances between the VoCs and the sera of vaccinated or challenged hamsters, we obtained sera obtained from hamsters ($N = 6$ per virus) challenged with the ancestral virus, Alpha, Beta, Gamma, Delta, Kappa, or Omicron VoC at 28 days post inoculation[25]. Live VN titers were assessed against the same seven VoCs and as shown previously[26], antigenic cartography showed that most VoCs cluster together, but the Omicron VoC is an outlier. Furthermore, within the cluster of VoCs, the Beta and Delta VoCs are furthest apart (Supplementary Fig. 2A). We then used antigenic cartography on the live VN titers of vaccinated hamster sera obtained against the ancestral, Beta, and Delta VoCs. A clear segregation by vaccine regimen was observed: sera obtained from hamsters which received AZD2816 only was strongly biased towards the Beta VoC antigen, whereas sera obtained from hamsters which received AZD1222 followed by AZD2816 had better reactivity to all antigens but was more biased towards ancestral antigen (Supplementary Fig. 2B).

### AZD2816 vaccination protects hamster against Beta VoC challenge

On Day 0, 18 animals per group (9 M, 9 F) were challenged via the intranasal route with SARS-CoV-2 Beta VoC. Controls showed weight loss starting at day 5 and full recovery at day 12–14. In contrast, vaccinated hamsters maintained weight during the experiment (Fig. 2A). Significant differences were observed between relative weights of the control and vaccinated groups on the day of peak weight loss compared to day 0 (Day 7, Fig. 2B, $p = 0.0034$ and 0.0137, Kruskall Wallis test). On day 3 and 5, six animals per group were euthanized and lung tissue was collected. In lung tissue of controls, subgenomic viral RNA

(sgRNA) was high on both days (12/12 lungs positive, median of $3.6 \times 10^9$ and $1.6 \times 10^{10}$ copies/gram, respectively). In contrast, no or reduced sgRNA was detected in lung tissue collected from prime only animals (1/6 and 0/6 lungs positive on day 3 and 5 respectively, significant by Kruskall Wallis test). Likewise, sgRNA in lung tissue collected on day 5 from prime-boost animals was low (1/6 lungs positive, significant by Kruskall Wallis test), whereas 2/6 lungs collected on day 3 from the prime-boost group were found to be positive at levels equivalent to the control group (no significance, Fig. 2C). We then investigated the presence of infectious virus particles in lung tissue and found a similar pattern to sgRNA detection (Fig. 2C). Oropharyngeal swabs were collected on day 1, 3, 5, and 7, and the presence of sgRNA was investigated. No reduction of sgRNA load in swabs was detected on day 1 in vaccinated groups compared to the control. However, a significant reduction in sgRNA load was found on day 3 and 5 in prime only animals. On day 7, both vaccinated groups showed a reduction in sgRNA load in oropharyngeal swabs, showing that although vaccination did not prevent infection, it did significantly reduce the window of shedding (Fig. 2D). We then investigated the presence of infectious virus in swabs obtained on day 3 and 5. A significant difference was found between the amount of infectious virus present in swabs obtained from vaccinated animals compared to control animals on both days (Fig. 2D).

Area-under-the-curve analysis, used as a measurement of total amount of sgRNA detected in oropharyngeal swabs throughout the experiment, showed a significant reduction in the prime-only group, but not the prime-boost group, compared to controls (Fig. 2E).

Serum was collected from all hamsters on day 5 post-challenge, and antibody titers were determined via ELISA as well as pseudovirus and live VN assays as described above (Supplementary Fig. 3). A

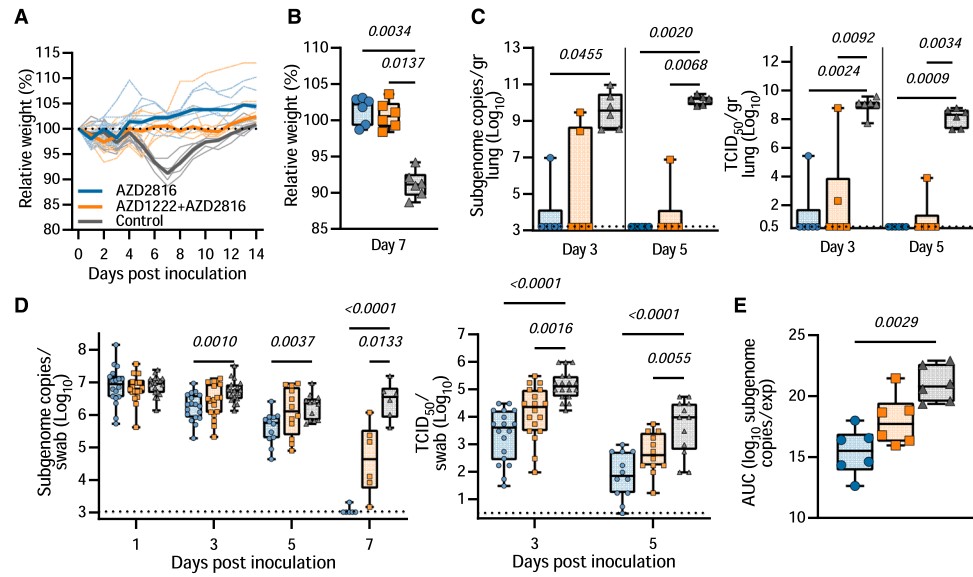

**Fig. 2 | Vaccination of Syrian hamsters with AZD2816 or AZD1222 followed by AZD2816 reduces lower respiratory tract infection by the Beta VoC.** Twenty-eight days post final vaccination, hamsters were challenged with $10^4$ TCID$_{50}$ of the Beta variant, via the intranasal route. **A** Relative weight in comparison to day 0. Thick lines = median weight per group, thin lines = individual animals, dotted line = 100% relative weight. Days 0–3 $N = 18$, Days 4–5, $N = 12$, Days 6–14 $N = 6$. **B** Boxplot (minimum to maximum) of relative weight at day 7. Statistical significance was determined via a Kruskall-Wallis test followed by Dunn's multiple comparisons test. **C** Boxplot (minimum to maximum) of subgenomic SARS-CoV-2 RNA (sgRNA) (left panel) and infectious virus isolation (right panel) in lung tissue harvested on day 3 and 5 ($N = 6$). Statistical significance was determined via a Kruskall Wallis test followed by Dunn's multiple comparisons test. Dotted line = limit of detection. **D** Boxplot (minimum to maximum) of sgRNA (left panel) and

infectious virus (right panel) in oropharyngeal swabs taken on day 1 (sgRNA only, $N = 18$), 3 ($N = 18$), 5 ($N = 12$), and 7 (sgRNA only, $N = 6$). Statistical significance was determined via a mixed-effects analysis followed by Dunnett's multiple comparisons test, comparing vaccinated groups against control group. Dotted line = limit of detection. **E** Boxplot (minimum to maximum) of the area under the curve (AUC) analysis of shedding as measured by sgRNA analysis in swabs collected on 1–7 days post-inoculation. Statistical significance was determined via a Kruskall-Wallis test followed by Dunn's multiple comparisons test. $N = 6$, circles = hamsters vaccinated with AZD2816, squares = hamsters vaccinated with AZD1222 followed by AZD2816, triangles = hamsters vaccinated with ChAdOx1 GFP. All boxplots are drawn from first quartile to third quartile, with a line at the median. Whiskers go from each quartile to minimum or maximum values. Source data are provided as a Source Data file.

significantly higher binding antibody titer against Beta S compared to ancestral and Delta S was detected in both vaccine groups via ELISA. In live VN assays, higher VN titers were found against Beta compared to Delta VoC in the prime only group. No differences between live VN titers against variants were found in the prime-boost group. In the pseudotype VN assay, titers against Omicron were lower than against ancestral, Beta, and Delta. Furthermore, titers against K417N S were higher than ancestral S. (Supplementary Figs. 1B and 3). We then investigated the correlation between sgRNA load in swabs on the day of necropsy with corresponding antibody levels. A significant correlation could be found between live VN titer against ancestral virus, the Beta, and Delta VoC, and sgRNA in swabs, but not between binding antibodies or pseudotype VN titer against any S variant and sgRNA in swabs (Supplementary Fig. 4). Antigenic cartography did not show a shift in the positioning of the hamster sera in relationship to the antigens, suggesting that the challenge with the Beta VoC did not significantly change the specificity of the humoral response five days post-challenge (Supplementary Fig. 2C).

Lung pathology was scored by a board-certified veterinary pathologist blinded to study groups (Fig. 3). SARS-CoV-2-related pathology was observed in all animals of the control group. On day 3 post Beta challenge, minimal-to-moderate acute bronchiolitis was observed affecting less than 1% of the lung. Histological lesions consisted of a moderate subacute broncho-interstitial pneumonia affecting between 30–50% of pulmonary tissue. Lesions were characterized by broncho-interstitial pneumonia centered on terminal bronchioles and extending into the adjacent alveoli. Alveolar septa were expanded by edema fluid and leucocytes. SARS-CoV-2 antigen staining was numerous within bronchiolar epithelium on day 3, whereas this had mostly moved to type I and II pneumocytes on day 5. In contrast, antigen staining in the vaccinated groups was lower compared to the control groups; lung samples obtained from the prime-only group were negative, whereas antigen staining in lung samples obtained from the prime-boost group was scored between none to moderate (Fig. 3, Supplementary Table 1, Supplementary Fig. 5).

### AZD2816 vaccination protects hamster against Delta VoC challenge

To investigate the efficacy of a vaccine optimized for the Beta S against the Delta variant, 18 hamsters per group (9 M, 9 F) were challenged with the Delta VoC via the intranasal route. As observed upon challenge with the Beta variant, vaccinated animals did not show any weight loss throughout the experiment, whereas control animals did lose weight (Fig. 4A). Indeed, differences in weight loss between the control and vaccinated groups were significant on day 7 (Fig. 4B). High levels of sgRNA could be detected on both day 3 and 5 in lung tissue of control animals (12/12 lungs positive, median of $5.6 \times 10^9$ and $5.0 \times 10^9$ copies/gram, respectively). In contrast, the majority of lung tissue obtained from vaccinated animals was negative for sgRNA, viral sgRNA was detected in 1/6 lungs on each day for the prime-boost group, versus 2/6 on day 3 and 0/6 on day 5 for the prime-only group (Fig. 4C). No infectious virus was detected in the lungs of vaccinated animals on day 5 and was limited on day 3. This is in contrast to control lungs, in which high titers of infectious virus were detected in all samples (Fig. 4C).

Significant differences in sgRNA and infectious virus detected in oropharyngeal swabs were limited to day 3 and day 5 compared to controls (Fig. 4D). In contrast to what we observed in animals inoculated with the Beta variant, we did not see a decrease in the window of shedding of vaccinated animals compared to control animals. Area-under-the-curve analysis (as a measurement of total amount of sgRNA detected in oropharyngeal swabs throughout the experiment) showed that animals that received a prime only vaccination shed significantly less than control animals (Fig. 4E).

In sera collected on day 5 post-challenge, a significantly higher binding antibody titer against Delta S compared to ancestral S was detected in the prime-only group (Supplementary Fig. 3). In the live VN assay, higher VN titers were found against the Beta VoC compared to ancestral virus in the prime-only group (Supplementary Fig. 3). Significant differences in the pseudotype VN assay were found between Omicron and Beta as well as Delta VoCs (Supplementary Fig. 3), as well as between ancestral and E484K S in the prime-only group, and ancestral and K417N S in the prime-boost group. (Supplementary Fig. 1C). A linear correlation was found between sgRNA in swabs and correlating binding antibodies against ancestral, Beta, and Delta S. For live VN titers, a significant correlation was found with the Beta and Delta, but not ancestral, VoCs (Supplementary Fig. 6). As with the Beta challenge, antigenic cartography did not show a shift in the positioning of the hamster sera in relationship to the antigens at 5 days post-challenge, as evident by the overlapping positioning of the sera from the animals challenged with Beta or Delta (Supplementary Fig. 2C).

SARS-CoV-2-related pathology post-Delta challenge was observed in all animals of the control group and did not differ from animals challenged with the Beta VoC (Fig. 5, Supplementary Table 1, Supplementary Fig. 5). In the prime-only group, a minimal-to-moderate bronchiolitis was observed in some animals on day 3. In the prime-boost group, bronchiolitis was either absent or minimal. This was combined with reduced antigen staining in the bronchiolar epithelium in both groups compared to controls. No pathology or antigen staining was observed on day 5, except for one animal in the prime-boost group, which is the only animal that was positive for sgRNA at this time point.

### AZD1222 and AZD2816 vaccination protects hamster against Omicron VoC challenge

We then investigated the efficacy of AZD1222 and AZD2816 against the Omicron BA1. VoC. Here, groups of hamsters were vaccinated with a single dose of AZD1222, AZD2816, or ChAdOx1 GFP, and serum was collected 14 days post-vaccination (Fig. 6A). Binding antibodies against different S proteins were detected using the Mesoscale V-PLEX SARS-CoV-2 panel 23, in-house optimized for hamster sera. Upon vaccination with AZD1222, binding antibodies were highest for ancestral and Alpha S and lowest for Omicron. Upon vaccination with AZD2816, antibody levels were similar for ancestral, Alpha, Beta, and Gamma S, but dropped for Delta and Omicron S (Fig. 6B). Live VN titers against the Omicron VoC were significantly lower compared to the ancestral variant (Fig. 6C). 28 days after vaccination, animals were challenged with the ancestral variant or Omicron VoC. One animal in the ChAdOx1 GFP vaccine group to be challenged with ancestral virus was euthanized prior to challenge due to complications unrelated to vaccine administration. As previously reported[27], we did not see weight loss in control hamsters challenged with the Omicron VoC, whereas this weight loss was present in control hamsters challenged with ancestral virus (Fig. 6D). Four animals per group were euthanized on day 3 and day 5. No significant differences in lung: body weight ratio were observed, although lung:body weight ratio was relatively high on day 5 for control hamsters inoculated with ancestral virus (Fig. 6E). As expected, AZD1222 vaccination resulted in significantly reduced viral genome copies and infectious virus in lung tissue (Fig. 6F). However, replication of the Omicron VoC in hamster lung tissue was low and although a reduction in virus in lung tissue of vaccinated hamsters was observed, particularly in the animals which received AZD2816, no significance was reached. Oropharyngeal swabs were obtained on days 1-5 and analyzed for sgRNA and infectious virus. Shedding of the control animals infected with the Omicron VoC was similar to control and vaccinated animals that were infected with the ancestral variant, albeit lower on day 1. However, vaccinated animals inoculated with the Omicron VoC had significantly lower shedding on day 3, day 4, and day 5 compared to controls (Fig. 6G). The amount of ancestral infectious

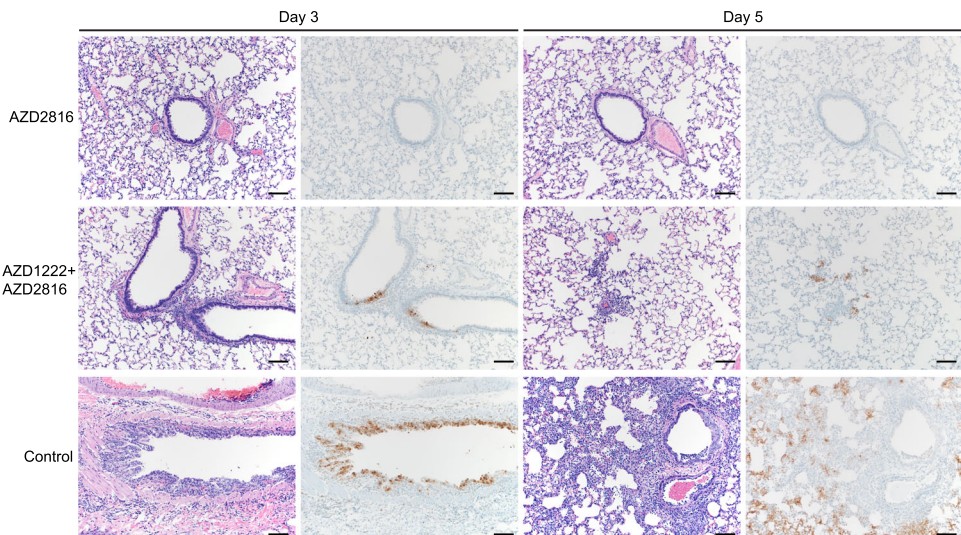

**Fig. 3 | Pulmonary effects of intranasal challenge with the Beta VoC in vaccinated and control hamsters at day 3 and 5 post challenge.** H&E staining (1st and 3rd column) and IHC staining against N protein (brown, 2nd and 4th column), 100x, scale bar = 100 μm. $N = 6$. No pathology nor antigen staining observed in animals which received an AZD2816 vaccination. No pathology observed in animals which received an AZD1222 + AZD2816 vaccination. Compared to control, limited staining of bronchiolar epithelium observed on days 3 and 5. Control animals show progression from bronchiolitis on day 3 to bronchointerstitial pneumonia on day 5, at which point alveolar septa are expanded by edema fluid and leucocytes. Staining of bronchiolar epithelial cells, type I&II pneumocytes, and rare macrophages. Images are representative of observations within 100% of a complete lung section containing all lobes.

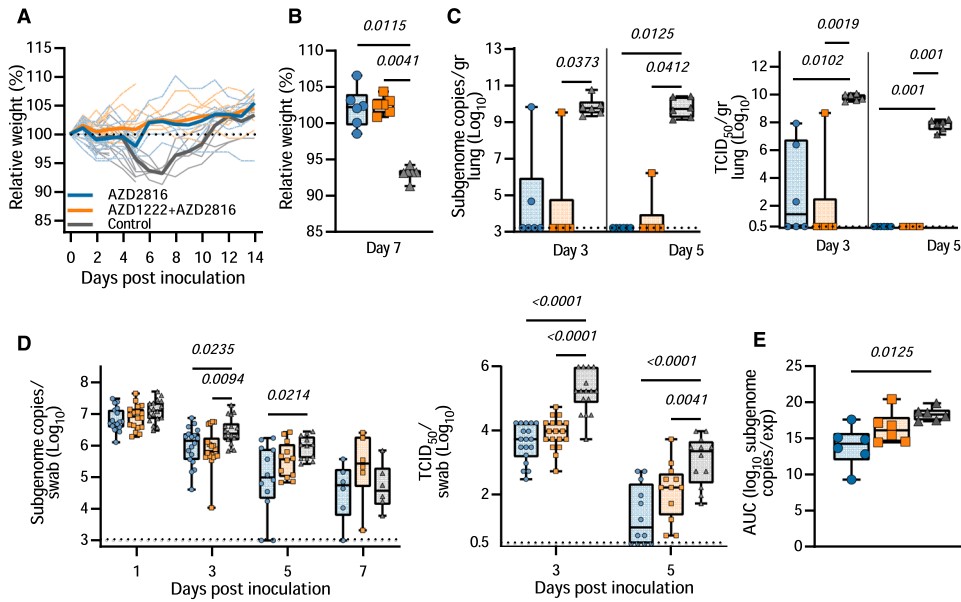

**Fig. 4 | Vaccination of Syrian hamsters with AZD2816 or AZD1222 followed by AZD2816 reduces lower respiratory tract infection by the Delta VoC.** Twenty-eight days post final vaccination, hamsters were challenged with $10^4$ TCID$_{50}$ of the Delta variant via the intranasal route. **A** Relative weight in comparison to day 0. Thick lines = median weight per group, thin lines = individual animals, dotted line = 100% relative weight. Days 0–3 $N = 18$, Days 4–5, $N = 12$, Days 6–14 $N = 6$.. **B** Boxplot (minimum to maximum) of relative weight at day 7. Statistical significance was determined via a Kruskall-Wallis test followed by Dunn's multiple comparisons test. **C** Boxplot (minimum to maximum) of sgRNA (left panel) and infectious virus isolation (right panel) in lung tissue harvested on days 3 and 5 ($N = 6$). Statistical significance was determined via a Kruskall-Wallis test followed by Dunn's multiple comparisons test. Dotted line = limit of detection. **D** Boxplot (minimum to maximum) of sgRNA (left panel) and infectious virus (right panel) in oropharyngeal swabs taken on day 1 (sgRNA only, $N = 18$), 3 ($N = 18$), 5 ($N = 12$), and 7 (sgRNA only, $N = 6$). Statistical significance was determined via a mixed-effects analysis followed by Dunnett's multiple comparisons test, comparing vaccinated groups against control group. Dotted line = limit of detection. **E** Boxplot (minimum to maximum) of the AUC analysis of shedding as measured by sgRNA analysis in swabs collected on 1–7 days post-inoculation. Statistical significance was determined via a Kruskall-Wallis test followed by Dunn's multiple comparisons test. $N = 6$, circles = hamsters vaccinated with AZD2816, squares = hamsters vaccinated with AZD1222 followed by AZD2816, triangles = hamsters vaccinated with ChAdOx1 GFP. All boxplots are drawn from first quartile to third quartile, with a line at the median. Whiskers go from each quartile to minimum or maximum values. Source data are provided as a Source Data file.

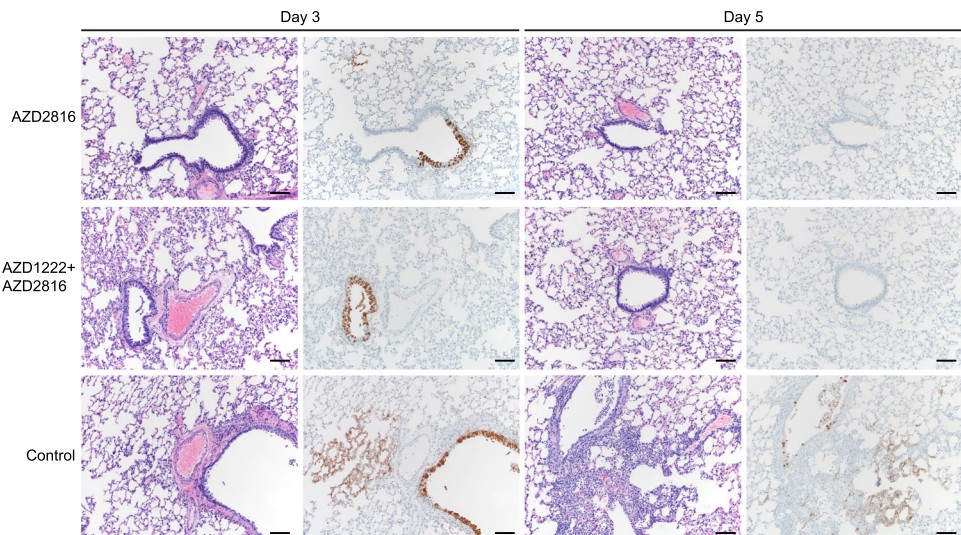

**Fig. 5 | Pulmonary effects of intranasal challenge with the Delta VoC in vaccinated and control hamsters at day 3 and 5 post challenge.** H&E staining (1st and 3rd column) and IHC staining against N protein (brown, 2nd and 4th column), 100x, scale bar = 100 μm. *N* = 6. Limited bronchiolitis with epithelial cell necrosis observed on day 3, which was resolved on day 5, in animals that received an AZD2816 vaccination. No pathology observed in animals which received an AZD1222 + AZD2816 vaccination. Compared to controls, limited staining of bronchiolar epithelium observed on day 3, which was resolved on day 5 in both vaccine groups. Control animals show progression from bronchiolitis on day 3 to bronchointerstitial pneumonia on day 5, at which point alveolar septa are expanded by edema fluid and leucocytes. Staining of bronchiolar epithelial cells, type I & II pneumocytes, and rare macrophages on both days. Images are representative of observations within 100% of a complete lung section containing all lobes.

virus shed was lower in vaccinated animals compared to control animals on day 3 (Fig. 6G). Area-under-the-curve analysis was performed on the four animals per group that were euthanized on day 5 as a measure of total amount of viral sgRNA shed throughout the experiment. Significantly lower shedding was detected in the vaccinated animals infected with the Omicron VoC, but not in those infected with the ancestral virus (Fig. 6H).

Lung pathology was scored by a board-certified veterinary pathologist blinded to study groups (Fig. 7, Supplementary Table 1, Supplementary Fig. 7). SARS-CoV-2-related pathology differed from what was observed in the animals inoculated with the Beta or Delta VoC. On day 3 and 5 post Omicron challenge, no evidence of pulmonary pathology was noted in the lower airway in any of the control animals, however minimal subacute inflammation and multifocal necrosis was noted in the trachea of 2/4 and 1/4 control animals on day 3 and 5, respectively. SARS-CoV-2 antigen staining was observed in 3/4 tracheal tissues on day 3. On day 5, rare staining was observed in bronchial and bronchiolar epithelium in 2/4 control animals, in type I and II pneumocytes in l/4 control animals, and in tracheal epithelium in 1/4 control animals. In animals vaccinated with AZD1222, 1/4 animals had a minimal subacute inflammation and multifocal necrosis of the tracheal epithelium. Rare SARS-CoV-2 antigen staining was observed in tracheal epithelium of 3/4 AZD1222 animals, and scattered SARS-CoV-2 antigen staining was observed in bronchial and bronchiolar epithelium of 1/4 AZD1222 animals. On day 5, no SARS-CoV-2 antigen staining is observed, and minimal interstitial pneumonia was seen in 1/4 AZD1222 animals. In animals vaccinated with AZD2816, no pathology was observed on day 3, and limited to minimal interstitial pneumonia in 1/4 AZD2816 animals on day 5. Antigen staining was limited to rare tracheal epithelium staining on day 3, with no staining observed on day 5 (Fig. 7, Supplementary Fig. 7).

## Discussion
We have previously shown that despite reduced neutralizing antibody titers in sera obtained from hamsters vaccinated with AZD1222 against the Beta VoC, hamsters were fully protected against lower respiratory tract infection[28]. Other vaccines have performed differently in animal models. Tostanoski et al. show that although vaccination with Ad26.COV2.S (an adenovirus-vectored vaccine encoding ancestral S protein) reduced the viral load detected in lung tissue of hamsters challenged with the Beta VoC at 14 days post-challenge, this difference was not significant for gRNA[29]. Likewise, rhesus macaques vaccinated with the same vaccine showed higher viral loads in bronchoalveolar lavage and nasal swabs when challenged with the Beta VoC compared to ancestral SARS-CoV-2[30]. Finally, Corbett et al. show that sgRNA was detected in bronchoalveolar lavage and nasal swabs from rhesus macaques vaccinated with the Moderna vaccine mRNA-1273 (encoding ancestral S) and challenged with the Beta VoC[31], whereas this was limited in rhesus macaques that were challenged with ancestral virus[32]. These data suggest that while vaccines which encode the ancestral spike can protect against hospitalization and death caused by VoC, a variant-specific vaccine may result in increased protection against disease and onward transmission.

Thus, we investigated the protective efficacy of the vaccine AZD2816, which encodes the S protein of the Beta VoC, in the hamster model. In contrast, to control animals, upon challenge with either the Beta, Delta, or Omicron VoC, little-to-no viral RNA was found at 5 days post-challenge in the lower respiratory tract of the vaccinated hamsters. Thus, the vaccine regimens utilized in the current study, including single-dose AZD2816, are protective against all three VoCs in the hamster model.

Vaccine and variant-specific differences were observed in the different experiments. In the Beta VoC study, lung tissue from 2/6 prime-boost vaccinated animals was positive for sgRNA at day 3, combined with higher antigen staining in this group compared to the prime-only group. Furthermore, whereas total shedding was reduced in the prime-only group compared to controls, this was not the case for the prime-boost group. This suggests that initial priming with one VoC S may shape the immune response to subsequent vaccinations. Indeed, our humoral immune response analysis showed higher titers for the Beta S and E484K mutation compared to ancestral or Delta S in the prime-only group, but not the prime-boost group. In contrast, in the Delta VoC study, the prime-boost group appeared to be slightly better protected than the prime-only group, mostly evident in

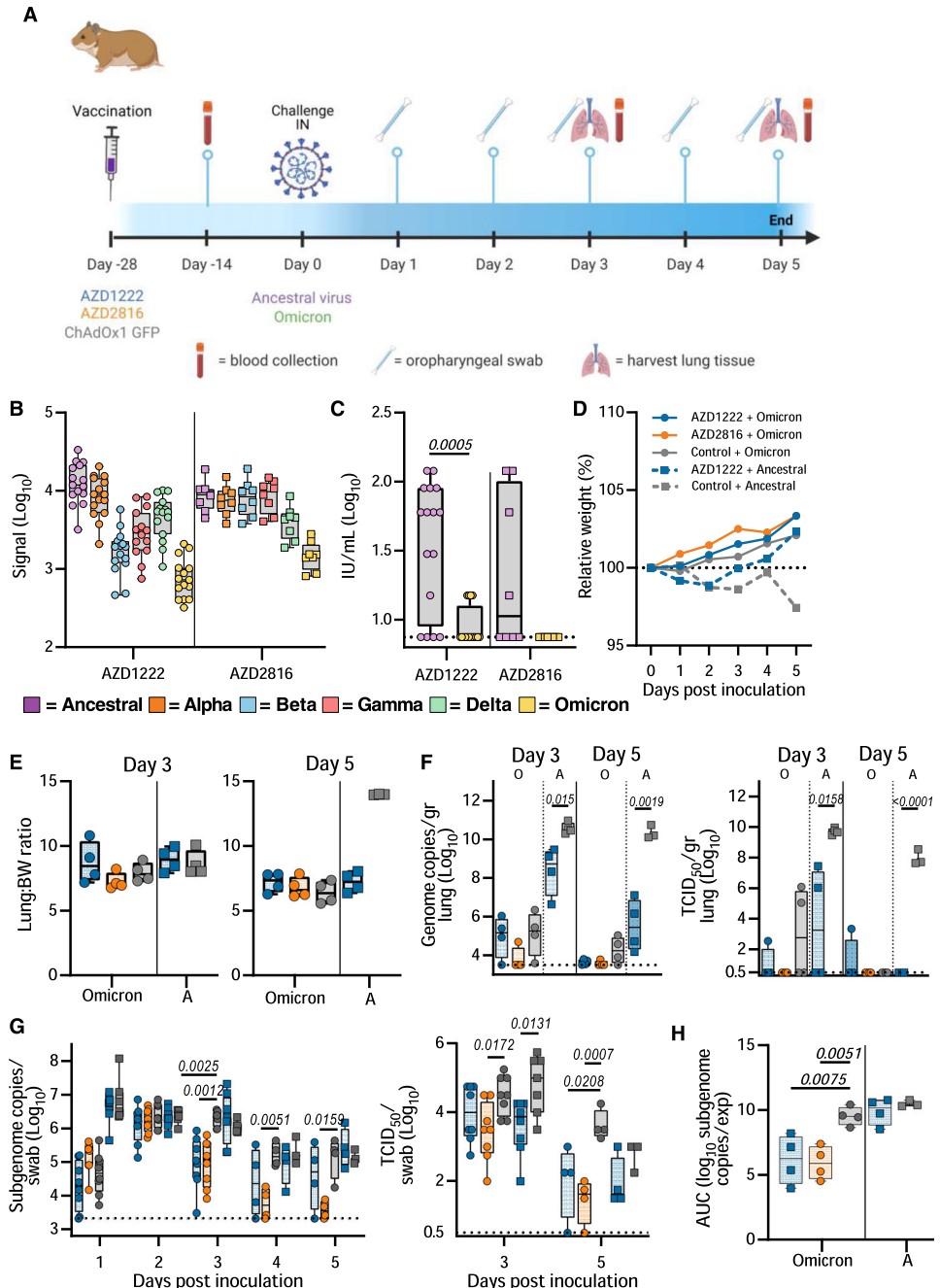

**Fig. 6 | Vaccination of Syrian hamsters with AZD2816 or AZD1222 reduces shedding by the Omicron VoC. A** Schematic overview of experiment. Hamsters were vaccinated with AZD1222, AZD2816, or ChAdOx1 GFP on day −28. Twenty-eight days post final vaccination, hamsters were challenged with $10^3$ TCID$_{50}$ of the Omicron or ancestral variant, via the intranasal route. Created with BioRender.com. **B** Boxplot of binding IgG antibody signal in hamster sera on day -14 against different SARS-CoV-2 S proteins. **C** Boxplot of virus neutralizing antibody titers in hamster sera obtained on day −14 against different ancestral virus or Omicron VoC. Statistical significance was determined via a Wilcoxon test. **B**, **C** Circles = hamsters vaccinated with AZD2816, squares = hamsters vaccinated with AZD2816. **D** Relative weight in comparison to day 0. Dotted line = 100% relative weight. $N = 8$ (Days 1–3) or 4 (Days 4–5). **E** Boxplot of lung:body weight ratio. **F** Boxplot of gRNA (left panel) and infectious virus (right panel) in lung tissue harvested on days 3 and 5 ($N = 4$).

Statistical significance was determined via a Kruskall-Wallis test followed by Dunn's multiple comparisons test. Dotted line = limit of detection. **G** Boxplot of sgRNA (left panel) and infectious virus (right panel) in oropharyngeal swabs taken on day 1–3 ($N = 8$), and 4–5 ($N = 4$). Statistical significance was determined via a mixed-effects analysis followed by Dunnett's multiple comparisons test, comparing vaccinated groups against control group. Dotted line = limit of detection. **H** Boxplot of the AUC analysis of shedding as measured by sgRNA analysis in swabs collected on 1–5 days post-inoculation. **C**–**H** Circles = hamsters challenged with Omicron, squares = hamsters challenged with ancestral variant. Statistical significance was determined via one-way ANOVA. $N = 5$. All boxplots are drawn from first quartile to third quartile, with a line at the median. Whiskers go from each quartile to minimum or maximum values. Source data are provided as a Source Data file.

pathology scoring. This may be due to the higher quantity of antibodies in the prime-boost group compared to the prime-only group.

We utilized antigenic cartography to study the positioning of individual sera of infected or vaccinated hamsters in relationship to the antigens. This analysis showed that sera obtained from animals challenged with a variant, grouped around the challenge virus, and that before Omicron, the Beta and Delta variant were furthest removed from each other. Sera obtained from hamsters vaccinated with

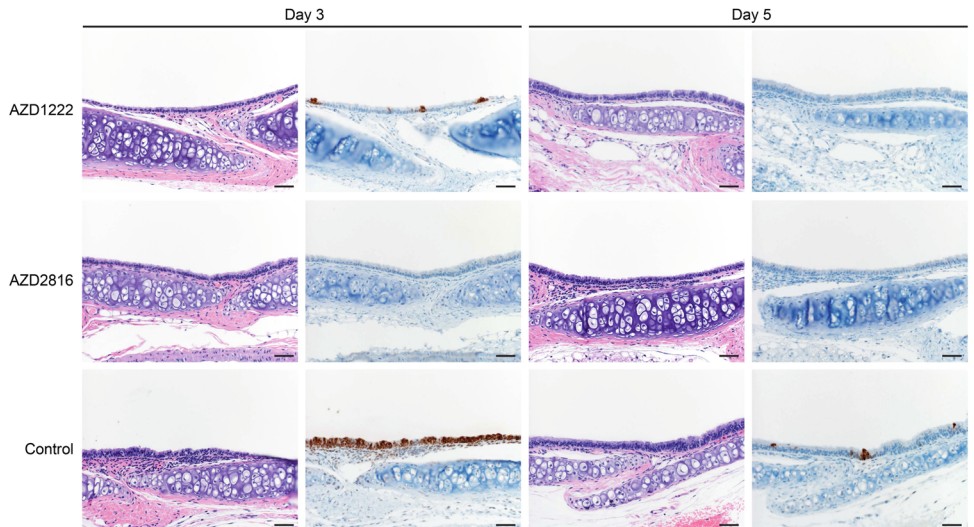

**Fig. 7 | Effects of intranasal challenge with the Omicron VoC on tracheal tissue in vaccinated and control hamsters at days 3 and 5 post challenge.** H&E staining (1st and 3rd column) and IHC staining against N protein (brown, 2nd and 4th column), 20x, scale bar = 50 μm. $N$ = 4. In animals that received an AZD1222 vaccination, minimal pathology in one animal and rare immunoreactivity in 3/4 was observed on day 3. No pathology nor antigen staining in animals on day 5. In animals that received an AZD2816 vaccination, no pathology or antigen staining was observed on day 3 or day 5. Control animals show moderate tracheitis on day 3 with moderate immunoreactivity. 2/4 animals had minimal pathology on day 5 and rare immunoreactivity.

AZD2816 grouped around the Beta variant, whereas sera that was obtained from animals vaccinated with AZD1222/AZD2816 was in between the ancestral and Beta variant. A subsequent challenge with either the Delta or Beta variant had not changed this positioning five days post challenge, supporting the notion that the responses are biased by initial exposure. A similar observation was shown in a recent preprint by Gagne et al., in which rhesus macaques received a regimen of mRNA-1273 against ancestral S, 3x, or mRNA-1273 2x and mRNA-Omicron as a booster. No difference in the immune response between the groups was observed, and both were protected against a challenge with Omicron[33]. Likewise, in a recent study utilizing a replicating RNA vaccine in hamsters, vaccination based on the Omicron VoC S protein resulted in the detection of neutralizing antibodies against Omicron. However, when hamsters received one or two vaccinations with self-replicating RNA vaccine based on ancestral S protein, followed by a boost with Omicron-specific vaccine, no neutralizing antibodies were detected[34]. Further research will be required to fully understand the role of original antigenic sin, and its importance in the human population.

A second explanation of the better protection against the Beta variant in single vaccinated animals would be the presence of vector immunity. Theoretically, the ability to neutralize the adenovector could reduce the efficacy of the booster. In this study, we did not collect sera in between vaccinations so were unable to investigate the presence of neutralizing antibodies against the vector after prime vaccination. In previous studies, we did show that ChAdOx1 neutralizing antibodies were present after a single vaccination, but that an increase in antibody titer was still observed upon boost[35]. In clinical studies, the presence of anti-vector immunity was shown after a two-dose regimen of ChAdOx1 nCoV-19, and did not prevent a third dose from boosting anti-SARS-CoV-2 responses[36]. Thus, even though anti-vector immunity can be detected, it is thought to have little effect on the ability to boost. Future studies are planned to investigate the importance of anti-vector immunity in the ChAdOx1 platform.

Similar to what has been reported by other groups, replication of the Omicron VoC was limited in the lower respiratory tract[27,37]. We were unable to detect any sgRNA, and gRNA was low compared to ancestral virus. However, using immunohistochemistry, we did observe a reduction in antigen staining in tracheal tissue in vaccinated animals. Despite the limited replication in the lower respiratory tract, the upper respiratory tract displayed much higher viral replication, comparable to ancestral virus. Both the AZD1222 and AZD2816 vaccine were able to reduce shedding within the Omicron-challenged groups suggesting that both vaccines are effective. This was recently confirmed by Gagne et al., who shows protection of NHPs against the Omicron VoC[33]. Further research is required to determine the extent in which the hamster model recapitulates human disease and infection kinetics with the Omicron VoC. We are adapting the model to increase replication in the lower respiratory tract by changing the inoculum dose as well as inoculation route.

Interestingly, whereas we previously reported on the lack of reduction in virus detected in oropharyngeal swabs when vaccines were given via the intramuscular route[28,38], vaccinated hamsters inoculated with the Omicron VoC displayed significantly reduced shedding of sgRNA compared to controls, even though shedding of control animals was at levels equal to animals inoculated with the ancestral virus. This difference was particularly evident in the AZD2816 group. Omicron has an E484A mutation in the S protein, whereas the Beta VoC has an E484K mutation. In our pseudovirus VN assays, we show a higher neutralization of pseudotypes with the E484K mutation compared to ancestral S in serum obtained from hamsters that only received the AZD2816 vaccination. It is possible that this also translates to the E484A mutation. However, we did not see an increase neutralization in live virus assays against the Omicron VoC in serum obtained from hamsters vaccinated with AZD2816 compared to those vaccinated with AZD1222. Further research is needed to determine whether the small difference observed between the two vaccines against the Omicron VoC is relevant, and why shedding is reduced. We currently do not know what explains the reduction in shedding for Omicron, but not ancestral virus. It is possible that the lack of virus replication in the lower respiratory tract plays a role. It should be noted that infectious virus was also reduced for ancestral virus on day 3 post-challenge.

A significant correlation was found between sgRNA load in oropharyngeal swabs and neutralizing antibody titers in animals, whereas binding antibodies titers were also predictors for sgRNA load in swabs in animals challenged with the Delta VoC. This finding confirms

previous reports of a correlation between binding and neutralizing antibodies and viral load in both hamsters[39] and non-human primates[31,40].

Our study confirms that AZD2816 is immunogenic in the hamster model and protects against infection of the lower respiratory tract with the Beta and Delta VoC, and the upper respiratory tract with Omicron VoC. Likewise, a single dose of AZD1222 protects against the Omicron VoC. Furthermore, initial immunization with AZD1222 followed by immunization with AZD2816 results in full protection against the Beta and Delta VoCs, and we predict it will also protect against Omicron. This confirms previous reports that a full antigenic match between the vaccine and the challenged virus is not required for protection of the lower respiratory tract.

## Methods

### Ethics
Animal experiments were conducted in an AAALAC International-accredited facility and were approved by the Rocky Mountain Laboratories Institutional Care and Use Committee following the guidelines put forth in the Guide for the Care and Use of Laboratory Animals 8th edition, the Animal Welfare Act, United States Department of Agriculture and the United States Public Health Service Policy on the Humane Care and Use of Laboratory Animals (protocol 2021-024E). The Institutional Biosafety Committee (IBC) approved work with infectious SARS-CoV-2 virus strains under BSL3 conditions. Virus inactivation of all samples was performed according to IBC-approved standard operating procedures for the removal of specimens from high containment areas.

### Cells and virus
The ancestral SARS-CoV-2 variant (USA/WA-CDC-WA1/2020, MN985325) was obtained from Natalie Thornburg and Sue Tong at the Centers for Disease Control and Prevention, Atlanta. SARS-CoV-2 variant B.1.1.7 (Alpha, England/204820464/2020, EPI_ISL_683466) was obtained from BEI resources. SARS-CoV-2 variant B.1.351 (Beta, USA/MD-HP01542/2021, EPI_ISL_890360) was obtained from Andrew Pekosz at John Hopkins Bloomberg School of Public Health. SARS-CoV-2 variant P.1 (Gamma, USA/GA-EHC-2811C/2021, EPI_ISL_7171744) was obtained from BEI resources. SARS-CoV-2 variant B.1.617.1 (Kappa, USA/ CA-SU-15_S02/2021, EPI_ISL_1675223) was obtained from BEI resources. SARS-CoV-2 variant B.1.617.2 (Delta, hCoV-19/USA/KY-CDC-2-4242084/2021) was obtained from BEI resources. SARS-CoV-2 variant B.1.1.529 (hCoV-19/USA/GA-EHC-2811C/2021, EPI_ISL_7171744) was obtained from Mehul Suthar, Emory University. All virus stocks were sequenced and analyzed using Bowtie2 version 2.2.9, and no SNPs compared to the patient sample sequence were detected. Virus propagation was performed in VeroE6 cells in DMEM (Gibco) supplemented with 2% fetal bovine serum (Gibco), 1 mM L-glutamine (Gibco), 50 U/ml penicillin (Gibco), and 50 μg/ml streptomycin (Gibco) (DMEM2). VeroE6 cells were maintained in DMEM supplemented with 10% fetal bovine serum, 1 mM L-glutamine, 50 U/ml penicillin, and 50 μg/ml streptomycin. No mycoplasma was detected in cells or virus stocks.

### Animal Experiment Beta and Delta challenge
ChAdOx1 nCoV-19 was formulated as previously described[35]. 44 (18 M, 26 F) 4-6-week-old Syrian hamsters (Envigo Indianapolis) were vaccinated with $2.5 \times 10^8$ infectious units of AZD1222, AZD2816, or ChAdOx1-GFP (approximately half the human vaccination dose) delivered intramuscularly in two 100 μL doses into the posterior thighs 56 or 28 days prior to challenge. Eight female animals were euthanized at 0 days post-inoculation and serum was collected. All animals were challenged intranasally with 40 μl containing $10^4$ TCID$_{50}$/mL virus in sterile DMEM (18 each (9 M, 9 F) for Beta and Delta variant). Body weights were recorded daily. Oropharyngeal swabs were collected in

1 mL of DMEM2. On day 3 and 5, 6 animals (3 M, 3 F) from each group were euthanized and lung samples were taken for qRT-PCR analysis, virus titrations, and histopathology. The remaining six animals in each group were monitored daily until day 21. One animal in the ChAdOx1 GFP vaccine group to be challenged with ancestral virus was euthanized prior to challenge due to complications unrelated to vaccine administration.

### Animal Experiment Omicron challenge
8 or 16 (1:1 ratio M/F) 4-6-week-old Syrian hamsters (Envigo Indianapolis) were vaccinated with $2.5 \times 10^8$ infectious units of AZD1222, AZD2816, or ChAdOx1-GFP delivered intramuscularly in two 100 μL doses into the posterior thighs 28 days prior to challenge. Serum was collected via retro-orbital bleed 14 days post-vaccination. 8 animals per group were challenged intranasally with 40 μL containing $10^3$ TCID$_{50}$/mL virus in sterile DMEM of Omicron or ancestral virus. Body weights were recorded daily. Oropharyngeal swabs were collected in 1 mL of DMEM2. On day 3 and 5, 4 animals (2 M, 2 F) from each group were euthanized and lung samples were taken for qRT-PCR analysis, virus titrations, and histopathology.

### Animal experiment VoC challenge for antigenic cartography
Sera was obtained from hamsters challenged as described previously[25]. Briefly, six 4-6-week-old Syrian hamsters (Envigo Indianapolis) were challenged intranasally with 40 μL containing $10^3$ TCID$_{50}$/mL virus in sterile DMEM. The VoCs used were: ancestral virus, Alpha, Beta, Gamma, Kappa, Delta, and Omicron. Twenty-eight days post-challenge, sera was collected for live VN assays.

### RNA extraction and quantitative reverse-transcription polymerase chain reaction
RNA was extracted from DMEM2 containing oropharyngeal swabs using the QiaAmp Viral RNA kit (Qiagen), and lung samples were homogenized and extracted using the RNeasy kit (Qiagen) according to the manufacturer's instructions and following high-containment laboratory protocols. Five μL of extracted RNA was tested with the Quantstudio 3 system (Thermofisher) according to the manufacturer's instructions using viral RNA-specific assays[41,42]. A standard curve was generated during each run using SARS-CoV-2 standards containing a known number of genome copies.

### Virus neutralization
Sera were heat-inactivated (30 min, 56 °C). After an initial 1:10 dilution of the sera, two-fold serial dilutions were prepared in DMEM2. 100 TCID$_{50}$ of SARS-CoV-2 was added to the diluted sera. After a 60 min incubation at 37 °C and 5% CO$_2$, the virus-serum mixture was added to VeroE6 cells and cells were further incubated for 6 days before assessment of CPE. The virus neutralization titer (ND100%) was expressed as the reciprocal value of the highest dilution of the serum that still inhibited virus replication. Three different positive serum controls were done next to NIBSC sera sample 20/130 by three different technicians, to determine IU/mL equivalent. NIBSC sera readout was 640-1066, compared to reported value at 1300 (1.5x higher). All serum samples were subsequently accompanied by positive controls on the plate. Assays were only approved if positive controls fell within the range previously determined by three technicians. Values were then multiplied by 1.5 to determine IU/mL.

### Generating lentiviral based pseudotypes bearing the SARS-CoV-2 S protein
Lentiviral-based SARS-CoV-2 pseudotyped viruses were generated in HEK293T cells incubated at 37 °C, 5% CO$_2$ as previously described[43]. Briefly, mutant SARS-CoV-2 expression plasmids (Clade A, Beta, Delta, Omicron, N501Y, E484K, K417N, L452R) were generated by site-directed mutagenesis or using the QuikChange Lightning Multi Site-

Directed Mutagenesis Kit (Agilent). All SARS-CoV-2 spike expression plasmids were based on the Wuhan-hu-1 reference sequence[44], with the additional substitutions D614G (except for clade A) and K1255*STOP (aka the Δ19 mutation or cytoplasmic tail truncation). Briefly, HEK293T cells were transfected with SARS-CoV-2 spike, along with the lentiviral plasmids p8.91 (encoding for HIV-1 gag-pol) and CSFLW (lentivirus backbone expressing a firefly luciferase reporter gene) with PEI (1 μg/mL) transfection reagent. Supernatants containing pseudo-typed SARS-CoV-2 were harvested and pooled at 48 and 72 h post transfection, centrifuged at 1300 x $g$ for 10 minutes at 4 °C to remove cellular debris and stored at −80 °C. SARS-CoV-2 pseudoparticles were titrated on HEK293T cells stably expressing human ACE2 and infectivity assessed by measuring luciferase luminescence after the addition of Bright-Glo luciferase reagent (Promega) and read on a GloMax Multi + Detection System (Promega).

### Micro neutralization test (mVNT) using SARS-CoV-2 pseudoparticles

Sera was diluted 1:20 in serum-free media in a 96-well plate in triplicate and titrated 3-fold. A fixed volume of SARS-CoV-2 pseudoparticles were added at a dilution equivalent to $10^5$ signal luciferase units in 50 μL DMEM-10% and incubated with sera for 1 h at 37 °C, 5% $CO_2$ (giving a final sera dilution of 1:40). Target cells stably expressing human ACE2 were then added at a density of $2 \times 10^4$ in 100 μL and incubated at 37 °C, 5% $CO_2$ for 48 h. Firefly luciferase activity was then measured after the addition of Bright-Glo luciferase reagent on a Glomax-Multi+ Detection System (Promega). CSV files were exported for analysis. Pseudotyped virus neutralization titers were calculated by interpolating the dilution at which a 50% reduction in reduction in luciferase activity was observed, relative to untreated controls, neutralization dose 50% (ND50).

### Enzyme-linked immunosorbent assay

MaxiSorp plates (Nunc) were coated with 100 ng (2 μg/ml) whole spike protein diluted in PBS for overnight adsorption at 4 °C. Plates were washed in PBS/Tween (0.05% v/v) and wells blocked using casein (ThermoFisher Scientific) for at least 1 h at RT. Standard positive sera (pool of hamster serum from AZD1222-AZD2816 vaccinated animals with high endpoint titer against original spike protein), individual hamster serum, negative and internal control samples were added to plates and incubated for at least 2 h at RT. Following washing, bound antibodies were detected by addition of Alkaline Phosphatase-conjugated goat anti-hamster IgG (Sigma-Aldrich, SAB3700455) (1:1000 dilution) for 1 h at RT and addition of p-Nitrophenyl Phosphate, Disodium Salt substrate (Sigma-Aldrich) and optimal density reading at 405 nm. An arbitrary number of ELISA units (EU) were assigned to the reference pool and optical density values of each dilution were fitted to a 4-parameter logistic curve using SOFTmax PRO software. ELISA units were calculated for each sample using the optical density values of the sample and the parameters of the standard curve. All data was log-transformed for presentation and statistical analyses.

### Binding antibody titers against different spike proteins on the Meso Quickplex

The V-PLEX SARS-CoV-2 Panel 13 (IgG) kit (MSD, K15463U) was used to run the hamster serum samples on the Meso Quickplex (MSD, K15203D). The 96-well plate was incubated with 150 μL of Blocker A solution at room temperature with shaking for 30 min, then washed 3 times with 150 μL/well of MSD Wash buffer. 50 μL of the standard curve and hamster serum samples were transferred to the plate in duplicates. Vaccinated hamster serum samples were diluted 10,000x, and ChAdOx1 GFP-vaccinated hamster serum samples were diluted 1,000x. The plate was sealed with shaking at room temperature for 2 h, followed by 3 washes with 1X MSD Wash buffer. An in-house MSD GOLD SULFO-

TAG NHS-Ester (MSD, R31AA-2) conjugated goat anti-hamster IgG secondary antibody (Thermo Fischer, SA5-10284) was diluted 10,000x in diluent 100 and 50 μL was applied to each well of the plate. The plate was sealed with shaking at room temperature for 1 h. After incubation, the plate was washed with 1X MSD Wash buffer as before, and 150 μL of MSD Gold Read Buffer B was added per well. The plate was read immediately by the MSD instrument. Arbitrary units (AU) were assigned to the standard curve of pooled SARS-CoV-2-positive hamster sera, which was used on each plate. AU/mL were calculated using the MSD Workbench 4.0 software.

### Antigenic Cartography

Antigenic maps were constructed as previously described[45,46] using the antigenic cartography software from https://acmacs-web.antigenic-cartography.org. In brief, this approach to antigenic mapping uses multidimensional scaling to position antigens (viruses) and sera in a map to represent their antigenic relationships. The maps here relied on post-SARS-CoV-2 infection serology data and post-vaccination serology data of Syrian hamsters. The positions of antigens and sera were optimized in the map to minimize the error between the target distances set by the observed pairwise virus-serum combinations in the VN assay described above and the resulting computationally derived map. Maps were constructed in 2, 3, 4, and 5 dimensions to investigate the dimensionality of the antigenic relationships. Both the infected animal and vaccinated animal datasets were two-dimensional with only small improvements in residual mean squared error of the maps as map dimensionality increased.

### Histology and immunohistochemistry

Lungs were perfused with 10% neutral-buffered formalin and fixed for at least 8 days. Tissue was embedded in paraffin, processed using a VIP 6 Tissue-Tek (Sakura Finetek) tissue processor, and then embedded in Ultraffin paraffin polymer (Cancer Diagnostics). Sections of 5 μm were deparaffinized in xylene, passed through 100% ethanol, and rehydrated in tap water. Sections were stained with Harris hematoxylin (Cancer Diagnostics, no. SH3777), decolorized with 0.125% HCl/70% ethanol, blued in Pureview PH Blue (Cancer Diagnostics, no. 167020), counterstained with eosin 615 (Cancer Diagnostics, no. 16601), dehydrated, and mounted in Micromount (Leica, no. 3801731). An in-house–generated SARS-CoV-2 nucleocapsid protein rabbit antibody (GenScript) at a 1:1000 dilution was used to detect specific anti–SARS-CoV-2 immunoreactivity, carried out on a Discovery ULTRA automated staining instrument (Roche Tissue Diagnostics) with a Discovery ChromoMap DAB (Ventana Medical Systems) kit. The tissue slides were examined by a board-certified veterinary anatomic pathologist blinded to study group allocations. Scoring was done as follows. H&E; no lesions = 0; less than 1% = 0.5; minimal (1–10%) = 1; mild (11–25%) = 2; moderate (26–50%) = 3; marked (51–75%) = 4; severe (76–100%) = 5. IHC attachment; none = 0; less than 1% = 0.5; rare/few (1–10%) = 1; scattered (11–25%) = 2; moderate (26–50%) = 3; numerous (51–75%) = 4; diffuse (76–100%) = 5.

### Statistical analyses

For all statistical analyses in this manuscript, we assumed a lack of Gaussian distribution and thus used non-parametric analyses. For repeated measurements on samples (e.g., ELISA against different antigens) we used the Friedman test followed by Dunn's multiple comparisons test comparing all groups. For independent sample analysis (e.g., viral RNA load in lung tissue from different vaccine groups) we used the Kruskall-Wallis test followed by Dunn's multiple comparisons test comparing all groups. For independent sample analysis over multiple time points (e.g., viral RNA load in swabs obtained over multiple days) we used a mixed-effects model with the Geisser-Greenhouse correction, followed by Dunnett's multiple comparisons tests comparing vaccinated groups against controls. To

investigate correlation between viral RNA load in swabs and humoral values, we performed a simple linear regression.

### Reporting summary

Further information on research design is available in the Nature Research Reporting Summary linked to this article.

## Data availability

The processed data are available at Figshare (10.6084/m9.figshare.19775650) and are provided in the Source Data file. Source data are provided with this paper.

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

## Acknowledgements

We would like to thank Mehul Suthar, Kathleen Cordova, Brian Smith, Jade Riopelle, Julia Port, Shane Gallogly, Julie Callison, Lara Myers, Nicolette Arndt, Trenton Bushmaker, Linda Couey, Brian Mosbrucker, Amanda Weidow, Natalie Thornburg, Sue Tong, Ranjan Mukul, Brandi Williamson, Myndi Holbrook, Emmie de Wit, Kyle Rosenke, Meaghan Flagg, Matthew Lewis, Craig Martens, Kent Barbian, Stacey Ricklefs, Sarah Anzick, Andrew Pekosz, Bin Zhou, Sujatha Rashid, Kimberly Stemple, Alan Sutherland, Anita Mora, and the animal care takers for their assistance during the study. Isolate hCoV-19/USA/MD-HP01542/2021 was obtained from Andrew Pekosz, John Hopkins Bloomberg School of Public Health. Isolate hCoV-19/USA/GA-EHC-2811C/2021 was obtained from Mehul Suthar, University Emory School of Medicine. The following reagent was deposited by the Centers for Disease Control and Prevention and obtained through BEI Resources, NIAID, NIH: hCoV-19/USA/KY-CDC-2-4242084/2021. This work was supported by the Intramural Research Program of the National Institute of Allergy and Infectious Diseases (NIAID), National Institutes of Health (NIH) (1ZIAAI001179-01, VJM), and AstraZeneca (SCG). The funders were not involved in study design, data collection or analysis.

## Author contributions

N.v.D and V.J.M. designed the studies, S.C.G. and T.L. provided the vaccine, N.v.D., J.S., D.R.A., T.A.S., R.J.F., C.K.Y., N.T., J.N., M.U., S.B.R., G.S., A.S., D.B., and V.J.M. performed the experiments, N.v.D., A.S., D.B., and C.A.R. analyzed results, N.v.D. and V.J.M. wrote the manuscript, all co-authors reviewed the manuscript.

## Funding

## Competing interests

S.C.G. is a co-founder and stock-holder of Vaccitech and named as an inventor on a patent covering the use of ChAdOx1-vector-based vaccines and a patent application covering a SARS-CoV-2 (nCoV-19) vaccine (UK patent application no. 2003670.3). T.L. is named as an inventor on a patent application covering a SARS-CoV-2 (nCoV-19) vaccine (UK patent application no. 2003670.3). The University of Oxford and Vaccitech, having joint rights in the vaccine, entered into a partnership with AstraZeneca in April 2020 for further development, large-scale manufacture and global supply of the vaccine. Equitable access to the vaccine is a key component of the partnership. Neither Oxford University nor Vaccitech will receive any royalties during the pandemic period or from any sales of the vaccine in developing countries. All other authors declare no competing interests.
