## [Peer review file · Nature Communications]

REVIEWER COMMENTS

Reviewer #1 (Remarks to the Author):

In this manuscript, the authors present AZD2816, a ChAdOx1-based vaccine which encodes the S protein of the SARS-CoV-2 Beta variant, either as single dose or after prime with AZ1222 (encoding the ancestral S) in Syrian hamsters. Analysis of binding and neutralizing antibodies includes a wide spectrum of variants and mutants, and the addition of Omicron variant data is timely and relevant. The AZD2816 vaccine candidate seemingly induces more protection against the Omicron variant in comparison to AZD1222, and may be a welcome addition to the arsenal of interventions to prevent SARS-CoV-2 infection and disease. A solid dataset is presented with large group sizes, clear figures and correct statistics. Some of the data, however, raise questions and are insufficiently addressed in the manuscript. Please find my comments below:

1) AZD1222+AZD2816-vaccinated hamsters seem to have higher viral RNA loads in the lungs and shed more virus over time in comparison to the AZD2816-vaccinated group. An explanation for this is given in lines 222-227, yet there is no mention on the possible influence of anti-vector (sterilizing) immunity that may preclude efficient boosting of anti-Spike immunity. Please address the possible interference of anti-ChAdOx immunity on the potency of boosting. Data showing the anti-ChAdOx Ab titers after priming and boosting would be informative to show possible presence of sterilizing anti-ChAdOx immunity after priming.

2) The authors have shown in previous publications that vaccination with AZD1222 reduces viral RNA to undetectable levels in the hamster model of SARS-CoV-2 infection. Figure 6F indicates however, that there is still a considerable amount of viral RNA (from the ancestral virus) that can be detected in hamster lungs after AZD1222 vaccination. Additionally, vaccination by AZD1222 seems to have no effect on viral shedding (figure 6G-H). This should be a reason for concern, but is only briefly mentioned in the results, and not discussed. Please discuss this difference in efficacy of AZD1222 in comparison to earlier reports.

Minor comments:

-In the results section (line 64) and methods section, please add the sex of the animals used in the study, as well as their numbers. Male Syrian hamsters generally show more severe disease characteristics after SARS-CoV-2 infection, so this information is not trivial and should be included.

-Figure 1 consists of data from 8 animals per group, when 18 animals were initially included in each group. Please indicate which animals were selected for the data of Figure 1, and how this selection was done.

-Figure 6: only 3 data points are shown at day 5 for the ChAdOx1 GFP control group, when all other groups included 4 animals. If an animal was excluded or died during the study, please indicate this in the methods section.

-The statistics used throughout the manuscript are clearly indicated in the figure legends, but the methods section still lacks a statistics paragraph. This should include all statistical tests used, and the reasoning to use each test for that particular type of data.

-Figure 6H: the data points on the right should have a grey color, not orange.

-Line 512: hamsters were vaccinated on day -28, not day 28.

-Line 201: delete comma at the end.

-Line 235: last word 'to' should be deleted.

Reviewer #2 (Remarks to the Author):

The manuscript describes immunization and challenge results with prime only or prime/boost schemes using an adenovirus vaccine vector expressing the ancestral spike protein or the VOC Beta spike.

The presented results are interesting, but there is several points which should be taken into consideration or which should be improved.

General points:

- The results section should be more structured, e.g. using sub-headings (e.g. Immunization and serological parameters, challenge infection VOC Beta etc.).
- Materials and methods are often repeated in the the results section. This should be reduced to the necessary minimum
- When „samples“ are mentioned it should be always clear which samples are meant (see e.g. comment line 94).
- A scoring of pathological analyses as well as of antigen staining on pathohistology would be very helpful and would improve the direct comparison of the different groups
- Is the Omicron challenge model in hamsters suitable to test vaccines when no moderate or severe disease is seen or the lower respiratory tract is not infected? Are the data of the upper respiratory tract helpful? A clear statement should be provided in the manuscript. What could be done for improved Omicron challenge models?

- Line 256 to 257: The statement is not correct for the Omicron VOC since there is no significant difference to the controls due to the restrictions of the Omicron model. The sentence has therefore to be adapted. It is also difficult to speak about „protection“ from Omicron infection (line 258 and also 260). The statements should be checked and adapted. What is protection against Omicron in the hamster model? Without a clear difference to the controls, the statements do not have the same value as for VOCs Beta or Delta.
- Please provide a short statement about the selection of the vaccine dosage in comparison to a human dosage.
- Please check important information provided like the used challenge dosage (see specific comments)
- Is there any virus isolation data available for the swab samples and tissue samples taken after challenge infection? These data would be helpful.
- The different serology assays should be differentiated in a clear way in the text to avoid any mix-up or misunderstandings (live VNT vs. infectious VNT etc.).
- Any explanation/description of the mentioned blinded pathology scoring is missing in the material and methods section. Both the methods as well as the scores should be provided. The heat map shows some data, but they are neither mentioned in the materials and methods section nor in the text (as scores).

Specific points:

- Line 74 to 78: This part is very difficult to follow. The structure of the role of VOCs or single mutations for the escape from neutralization should be improved to make it easier to follow the text.
- Line 82: why is here only the „binding“ mentioned. Is there also a biological consequence seen e.g. by reduced neutralization titers.
- Line 84: the term „animals were challenged“ is not very specified. The description should be here more precise (e.g. all animals? Where there also controls which were not challenged).
- Line 88: Why was day 3 post challenge infection not mentioned in the abstract?
- Line 91: for the information „limited sgRNA“ also the value range should be provided
- Line 94: the mentioned „samples“ should be specified
- Line 104: Why was serum detected on day 5 only? A later time point might have been more suitable. Please explain.
- Line 108: the titers should be presented for both the prime only group and the prime/boost group to allow direct comparison
- Line 112: the term „infectious VN titer“ is not very scientific and should be improved
- Line 122: Why is no dpi 3 data mentioned?

- Line 123: what means „relatively low“. Please try to explain in a more clear or quantified way. This is also true for „mostly negative“ or „none to moderate“. Please consider a scoring of lesions and staining results to make it more objective.
- Line 126: It is not clear from the text which groups were challenged (prime only or also prime/boost). Please clarify.
- Line 132 to 134: Please provide the genome loads also for the positive vaccinated animals
- Line 143: the term „live VN“ assays should be checked as well as the term „infectious VN assay“ (see also comment above).
- Line 153: it would be helpful to a scoring system of the pathological changes
- Line 166: The Omicron variant should be named more exactly (e.g. strain name and e.g. BA1 or BA2)
- Line 183: As mentioned above, it would be helpful to mention the scoring data also for the Omicron challenged animals
- Line 191-194: It is often not clear which „animals“ were meant. It would be helpful to name the groups more often and as defined as possible.
- Line 215: also the day 3 data should be discussed and should be compared to the day 5 data.
- Lines 220 to 227: It would be helpful to have some more explanations for the improved outcome of the prime only vaccination and to discuss this with more related references which might support the explanations given here. Is „imprinting“ something which could play a role here? Since this is a central part of the study, an improved and more detailed discussion should be provided.
- Lines 242-244: The unexpected reduction of viral replication in the upper respiratory tract in vaccinated animals following VOC Omicron challenge infection should be discussed in more detail. Why is there a difference in comparison to the other VOC data? Is there any other preprints/publications with similar or different data sets?
- Line 320 and line 127: please clarify the amount of challenge virus used (10e4/animal or 10e4/ml and 40µl per animal)
- Line 332: please provide a more detailed description of the determination of the virus neutralization titers. Is this e.g. ND50% or ND100% data?
- Line 380: How were the ELISA units calculated? What was the reference used for the standard curve? Please provide details.

Minor issues:

- Line 291: lung tissue ...“was“ positive
- Is there an official number for the ethics allowance for the trials? If yes, please provide.

Reviewer #3 (Remarks to the Author):

In this manuscript, the authors compare Adenovirus based vaccines containing the Spike antigen of ancestral or Beta variant SARS-CoV-2 in Syrian Golden Hamsters. A single vaccination with Beta-vaccine is compared to ancestral prime beta boost vaccination. Correlation with protection is investigated by challenging vaccinated animals with Beta and Delta SARS-CoV-2 variants of concern. Protection provided by single doses of both vaccines against the omicron variant was also tested. Vaccination resulted in protection as measured by reduction of lung virus titers (sgRNA) and body weight loss (for Ancestral, Beta and Delta VoC).

This study is very timely and addresses an important question: do booster vaccines need to be updated for variants of concern.

Fig. 1: I suggest to choose another acronym for ancestral virus instead of "A", since this may cause confusion with the Alpha variant.

The authors represent viral titers as subgenomic RNA copies in figures 2 and 4. Did authors also quantify replicating virus, for example by TCID50 or plaque assay?

Why is a different inoculum dose used for the first experiment and the second experiment that includes the omicron variant?

The high viral loads detected by oropharyngeal swabs compared to the lung virus titers and the effect of vaccination on them should be discussed in the context of transmission studies. Have the authors considered to test the same vaccination regimen on droplet transmission in a transmission study, given their expertise with these models?

The discussion would benefit if authors could elaborate more on how prior vaccination may affect the outcome of booster vaccination. Is this also seen in the clinic or in other preclinical models?

Dear reviewers,

We like to thank you for your time and your comments and suggestions have improved the manuscript considerably.

Please see a point-by-point response to the reviewers' concerns below.

Reviewer #1 (Remarks to the Author):

In this manuscript, the authors present AZD2816, a ChAdOx1-based vaccine which encodes the S protein of the SARS-CoV-2 Beta variant, either as single dose or after prime with AZ1222 (encoding the ancestral S) in Syrian hamsters. Analysis of binding and neutralizing antibodies includes a wide spectrum of variants and mutants, and the addition of Omicron variant data is timely and relevant. The AZD2816 vaccine candidate seemingly induces more protection against the Omicron variant in comparison to AZD1222, and may be a welcome addition to the arsenal of interventions to prevent SARS-CoV-2 infection and disease. A solid dataset is presented with large group sizes, clear figures and correct statistics. Some of the data, however, raise questions and are insufficiently addressed in the manuscript. Please find my comments below:

1) AZD1222+AZD2816-vaccinated hamsters seem to have higher viral RNA loads in the lungs and shed more virus over time in comparison to the AZD2816-vaccinated group. An explanation for this is given in lines 222-227, yet there is no mention on the possible influence of anti-vector (sterilizing) immunity that may preclude efficient boosting of anti-Spike immunity. Please address the possible interference of anti-ChAdOx immunity on the potency of boosting. Data showing the anti-ChAdOx Ab titers after priming and boosting would be informative to show possible presence of sterilizing anti-ChAdOx immunity after priming.

We thank the reviewer for their valuable comments. We agree with the notion that it would be nice to show anti-ChAdOx1 antibody titers between prime and boost. Unfortunately, our animals were not bled in between vaccinations (due to the experimental procedure of retro-orbital bleeds, we typically only do once on an animal per experiment). Therefore, we have chosen the following approach to address the reviewer's comments:

- 1. We have included an extra paragraph in the discussion, acknowledging the possibility that vector immunity may interfere, but also present evidence from previous manuscripts that the presence of vector immunity does not prevent the increase in antibody titer upon boosting, and thus is currently thought to be of minimal importance (Line 280-289).*
- 2. We have created antigenic maps, which provide a 2D image of the antibody response against different variants of concern elicited upon vaccination with either the AZD2816 vaccine alone, or the AZD1222 followed by the AZD2816 vaccine. This map clearly shows that AZD2816 alone preferentially neutralizes the Beta variant, whereas the AZD1222/AZD2816 falls in between the Beta and ancestral variant. We also included a secondary map showing antigenic cartography at 5 days post challenge with the Beta or Delta variant. Although antibody responses have increased (as evident by the antigens moving closer together, which symbolizes an increase in magnitude of the humoral response), the overall positioning of the sera is similar regardless of what virus the*

animals were challenged with. The map is discussed in the results (Line 91-101 and 138-140), discussion (Line 264-279), and methods (Line 486-496).

2) The authors have shown in previous publications that vaccination with AZD1222 reduces viral RNA to undetectable levels in the hamster model of SARS-CoV-2 infection. Figure 6F indicates however, that there is still a considerable amount of viral RNA (from the ancestral virus) that can be detected in hamster lungs after AZD1222 vaccination. Additionally, vaccination by AZD1222 seems to have no effect on viral shedding (figure 6G-H). This should be a reason for concern, but is only briefly mentioned in the results, and not discussed. Please discuss this difference in efficacy of AZD1222 in comparison to earlier reports.

We thank the reviewer for their comment. It should be noted that this is the first time that we show day 3 post challenge data, whereas previously we only showed day 5 data. When we compare the day 5 data shown in this manuscript to e.g. ‘Intranasal ChAdOx1 nCoV-19/AZD1222 vaccination reduces viral shedding after SARS-CoV-2 D614G challenge in preclinical models’ by van Doremalen et al., 2021, we do see some viral gRNA in lung tissue at 5 days post inoculation, similar to what is shown here. Importantly, sgRNA was only found in one out of four lung tissues at this timepoint in the current study.

Finally, the reviewer is correct about a lack of reduction in shedding in hamsters that received an IM vaccination. This has been reported by us previously (Intranasal ChAdOx1 nCoV-19/AZD1222 vaccination reduces viral shedding after SARS-CoV-2 D614G challenge in preclinical models’ by van Doremalen et al., 2021) and is typically seen by others with other vaccines as well (‘VSV-Based Vaccines Reduce Virus Shedding and Viral Load in Hamsters Infected with SARS-CoV-2 Variants of Concern’ by O’Donnell et al, 2022; ‘Adenovirus type 5 SARS-CoV-2 vaccines delivered orally or intranasally reduced disease severity and transmission in a hamster model’ by Langel et al, 2022). In addition, virus shedding is reported in vaccinated individuals (‘Duration of viral shedding and culture positivity with postvaccination SARS-CoV-2 delta variant infections’ by Siedner et al, 2022; ‘Duration of Infectious Virus Shedding by SARS-CoV-2 Omicron Variant-Infected Vaccinees’ by Takahashi et al, 2022). Although we do agree that this is of particular interest, it largely shows that the upper respiratory tract mucosal immunity plays a big role. Mucosal vaccines are one way we can hope to reduce this.

Minor comments:

-In the results section (line 64) and methods section, please add the sex of the animals used in the study, as well as their numbers. Male Syrian hamsters generally show more severe disease characteristics after SARS-CoV-2 infection, so this information is not trivial and should be included.

The specifics have now been provided throughout the result section. In our challenge experiments, we used males and females at a 1:1 ratio.

-Figure 1 consists of data from 8 animals per group, when 18 animals were initially included in each group. Please indicate which animals were selected for the data of Figure 1, and how this selection was done.

We apologize for the confusion on this section. Since we needed a relatively high amount of serum to be able to do the analyses we wanted to do at this time point, we had to euthanize 8 animals at 0 days post inoculation. This has now been clarified in the result and method sections.

-Figure 6: only 3 data points are shown at day 5 for the ChAdOx1 GFP control group, when all other groups included 4 animals. If an animal was excluded or died during the study, please indicate this in the methods section.

We thank the reviewer for this observation. Indeed, one animal was euthanized after vaccination but before challenge, due to issues unrelated to the study. This has now been stated in the methods and results section.

-The statistics used throughout the manuscript are clearly indicated in the figure legends, but the methods section still lacks a statistics paragraph. This should include all statistical tests used, and the reasoning to use each test for that particular type of data.

We thank the reviewer for this comment, we have included a statistical analyses section in the methods (Line 513-523).

-Figure 6H: the data points on the right should have a grey color, not orange.

-Line 512: hamsters were vaccinated on day -28, not day 28.

-Line 201: delete comma at the end.

-Line 235: last word 'to' should be deleted.

Many thanks for these observations, all have been corrected in the manuscript. We really appreciate your comments, and think they substantially improved the manuscript.

Reviewer #2 (Remarks to the Author):

The manuscript describes immunization and challenge results with prime only or prime/boost schemes using an adenovirus vaccine vector expressing the ancestral spike protein or the VOC Beta spike.

The presented results are interesting, but there is several points which should be taken into consideration or which should be improved.

General points:

- The results section should be more structured, e.g. using sub-headings (e.g. Immunization and serological parameters, challenge infection VOC Beta etc.).

We thank the reviewer for their comments and believe they have improved our manuscript. We have put sub-headings throughout the result section.

- Materials and methods are often repeated in the results section. This should be reduced to the necessary minimum

We have reduced certain aspects of the methods in the result section, whilst still honouring the changes that reviewer 1 requested.

- When „samples“ are mentioned it should be always clear which samples are meant (see e.g. comment line 94).

All use of the word ‘sample(s)’ throughout the manuscript has been revised and either replaced or clarified.

- A scoring of pathological analyses as well as of antigen staining on pathohistology would be very helpful and would improve the direct comparison of the different groups

The scoring of these samples for the Beta and Delta VoC challenge was performed and is represented in Extended Data Figure 4. We have now also included a table (Extended Data Table 1) including the Omicron VoC challenge data, so that the reader can easily compare the scores between groups.

- Is the Omicron challenge model in hamsters suitable to test vaccines when no moderate or severe disease is seen or the lower respiratory tract is not infected? Are the data of the upper respiratory tract helpful? A clear statement should be provided in the manuscript. What could be done for improved Omicron challenge models?

We thank the reviewer for their comment. Although we have not seen any replication in the distal end (alveoli and the type 1 and 2 pneumocytes), replication was still observed in the other parts of the lower respiratory tract. This is shown in Figure 7, where the vaccinated groups have an absence of omicron replication in the trachea. However, the absence of clear replication in the distal end of the lower respiratory tract can be overcome. If hamsters are inoculated via the intratracheal route instead of the intranasal route, replication in the lower respiratory tract is increased (see figure below, this is a separated study to determine the tropism changes of the Omicron variant in the hamster model). This would make it easier to investigate vaccine efficacy. We now allude to this work in the discussion (Line 298-300), and this data has been shared with the WHO and other platforms. We are currently planning to publish this data in a different manuscript. We do believe that the data from the upper respiratory tract is convincing in showing vaccine efficacy.

- Line 256 to 257: The statement is not correct for the Omicron VOC since there is no significant difference to the controls due to the restrictions of the Omicron model. The sentence has therefore to be adapted. It is also difficult to speak about „protection“ from Omicron infection (line 258 and also 260). The statements should be checked and adapted. What is protection against Omicron in the hamster model? Without a clear difference to the controls, the statements do not have the same value as for VOCs Beta or Delta.

We agree with the reviewer and have adapted the sentence to state that the vaccine protects against Beta and Delta in the lower respiratory tract, whereas protection against Omicron is seen in the upper respiratory tract. This is since we only looked at viral loads in the lungs and not other parts of the lower respiratory tract such as the trachea. The absence of replication by histopathology suggests an effect on the lower respiratory tract as well. This finding is now discussed in line 292-293.

- Please provide a short statement about the selection of the vaccine dosage in comparison to a human dosage.

A sentence has now been added to the methods section (Line 391)

- Please check important information provided like the used challenge dosage (see specific comments)

Changes have been made in the method section to clarify the challenge dose.

- Is there any virus isolation data available for the swab samples and tissue samples taken after challenge infection? These data would be helpful.

We thank the reviewer for their comment. We have now included titration data for all the swabs collected on 3- and 5-days post challenge, as well as infectious virus titers in all lung tissues. This data can be viewed in Figure 2, 4, and 6.

- The different serology assays should be differentiated in a clear way in the text to avoid any mix-up or misunderstandings (live VNT vs. infectious VNT etc.).

All instances of infectious VN have now been changed to live VN.

- Any explanation/description of the mentioned blinded pathology scoring is missing in the material and methods section. Both the methods as well as the scores should be provided. The heat map shows some data, but they are neither mentioned in the materials and methods section nor in the text (as scores).

I believe the reviewer may have missed this part in the methods. The following text is in the 'histology and immunohistochemistry' section (Line 497-512):

The tissue slides were examined by a board-certified veterinary anatomic pathologist blinded to study group allocations. Scoring was done as follows. H&E; no lesions = 0; less than 1% = 0.5; minimal (1-10%) = 1; mild (11-25%) = 2; moderate (26-50%) = 3; marked (51-75%) = 4; severe (76-100%) = 5. IHC attachment; none = 0; less than 1% = 0.5; rare/few (1-10%) = 1; scattered (11-25%) = 2; moderate (26-50%) = 3; numerous (51-75%) = 4; diffuse (76-100%) = 5.

Specific points:

- Line 74 to 78: This part is very difficult to follow. The structure of the role of VOCs or single mutations for the escape from neutralization should be improved to make it easier to follow the text.

We have now divided the text into paragraphs and adjusted the text so it is easier to follow.

- Line 82: why is here only the „binding“ mentioned. Is there also a biological consequence seen e.g. by reduced neutralization titers.

This has been changed to include neutralization.

- Line 84: the term „animals were challenged“ is not very specified. The description should be here more precise (e.g. all animals? Where there also controls which were not challenged).

The text has been adapted accordingly (Line 103).

- Line 88: Why was day 3 post challenge infection not mentioned in the abstract?

The abstract has now been adapted to include day 3.

- Line 91: for the information „limited sgRNA“ also the value range should be provided

For the control lung tissues, we give the median value. I chose not to do this for the vaccinated lung tissues, since these are all at the limit of detection. Additionally, providing the absolute value of the one sample that is positive will likely confuse the reader. I changed the word limited to reduced.

- Line 94: the mentioned „samples“ should be specified

This has been changed.

- Line 104: Why was serum detected on day 5 only? A later time point might have been more suitable. Please explain.

We agree with the reviewer, but unfortunately to reduce animal numbers this was not part of the experimental design and cannot be changed at this time. In future experiments, we will collect sera at a later timepoint.

- Line 108: the titers should be presented for both the prime only group and the prime/boost group to allow direct comparison

We have added a sentence that shows the lack of difference in VN response between variants in the prime-boost group (Line 131-132)

- Line 112: the term „infectious VN titer“ is not very scientific and should be improved

Infectious VN has now been changed to live VN

- Line 122: Why is no dpi 3 data mentioned?

Day 3 data was mentioned initially in line 121-122 which starts with the description of day 3 and finishes with day 5.

- Line 123: what means „relatively low“. Please try to explain in a more clear or quantified way. This is also true for „mostly negative“ or „none to moderate“. Please consider a scoring of lesions and staining results to make it more objective.

We agree with the reviewer that this was in need for some additional clarification. None to moderate refers to the pathology score as described in the method section and is the standard way of pathological scoring. We have changed the sentence to reflect this. Relatively low has been changed to lower, and mostly negative has been changed to negative.

- Line 126: It is not clear from the text which groups were challenged (prime only or also prime/boost). Please clarify.

The text has been changed to clarify this (Line 153-154).

- Line 132 to 134: Please provide the genome loads also for the positive vaccinated animals

As mentioned above, if we would put the median in as we do for the positive samples, this would be the limit of detection.

- Line 143: the term „live VN“ assays should be checked as well as the term „infectious VN assay“ (see also comment above).

This has been changed throughout.

- Line 153: it would be helpful to a scoring system of the pathological changes

This has been described in the methods. See previous comment.

The following text is in the ‘histology and immunohistochemistry’ section, (Line 497-512):

The tissue slides were examined by a board-certified veterinary anatomic pathologist blinded to study group allocations. Scoring was done as follows. H&E; no lesions = 0; less than 1% = 0.5; minimal (1-10%) = 1; mild (11-25%) = 2; moderate (26-50%) = 3; marked (51-75%) = 4; severe (76-100%) = 5. IHC attachment; none = 0; less than 1% = 0.5; rare/few (1-10%) = 1; scattered (11-25%) = 2; moderate (26-50%) = 3; numerous (51-75%) = 4; diffuse (76-100%) = 5.

- Line 166: The Omicron variant should be named more exactly (e.g. strain name and e.g BA1 or BA2)

We have now added BA.1 to the text, the strain name can be found in the methods section.

- Line 183: As mentioned above, it would be helpful to mention the scoring data also for the Omicron challenged animals

This has been described in the methods. See previous comments.

- Line 191-194: It is often not clear which „animals“ were meant. It would be helpful to name the groups more often and as defined as possible.

We have updated the manuscript and the experimental group has now been noted every time animals are mentioned.

- Line 215: also the day 3 data should be discussed and should be compared to the day 5 data. Day 3 data is discussed in the next paragraph ((Line 254-263)), since the data is more complex than the day 5 data.

- Lines 220 to 227: It would be helpful to have some more explanations for the improved outcome of the prime only vaccination and to discuss this with more related references which might support the explanations given here. Is „imprinting“ something which could play a role here? Since this is a central part of the study, an improved and more detailed discussion should be provided.

We agree with the reviewer. We have now included an antigenic map and discussed this further in the manuscript. Imprinting, or original antigenic sin, is indeed one of the possibilities we are considering. The study however was not set up to look into this, our main aim was to study the efficacy of the AZD2816 vaccination against VoCs. Thus, although the data suggests that OAS may be what plays a role here, it is not yet proven. We are currently working on follow-up studies to investigate this in more detail.

- Lines 242-244: The unexpected reduction of viral replication in the upper respiratory tract in vaccinated animals following VOC Omicron challenge infection should be discussed in more detail. Why is there a difference in comparison to the other VOC data? Is there any other preprints/publications with similar or different data sets?

As far as we know, the combination of lack of lower respiratory tract and reduction in upper respiratory tract upon vaccination in hamsters has not been reported. We have added a sentence in the discussion elaborating on this phenomenon.

- Line 320 and line 127: please clarify the amount of challenge virus used (10e4/animal or 10e4/ml and 40µl per animal)

The method sections now clearly states the dose per animal (Line 393-394).

- Line 332: please provide a more detailed description of the determination of the virus neutralization titers. Is this e.g. ND50% or ND100% data?

ND100% has now been added to the methods section (Line 425).

- Line 380: How were the ELISA units calculated? What was the reference used for the standard curve? Please provide details.

The calculation of the ELISA units were described in the methods section (Line 460-470), see below.

- *Standard positive sera (pool of hamster serum from AZD1222-AZD2816 vaccinated animals with high endpoint titer against original spike protein), individual hamster serum, negative and internal control samples were added to plates*
- *An arbitrary number of ELISA units (EU) were assigned to the reference pool and optical density values of each dilution were fitted to a 4-parameter logistic curve using SOFTmax PRO software. ELISA units were calculated for each sample using the optical density values of the sample and the parameters of the standard curve. All data was log-transformed for presentation and statistical analyses.*

Minor issues:

- Line 291: lung tissue ...“was“ positive

This has been adjusted.

- Is there an official number for the ethics allowance for the trials? If yes, please provide.

This number has been added to the method section (Line 368).

Reviewer #3 (Remarks to the Author):

In this manuscript, the authors compare Adenovirus based vaccines containing the Spike antigen of ancestral or Beta variant SARS-CoV-2 in Syrian Golden Hamsters. A single vaccination with Beta-vaccine is compared to ancestral prime beta boost vaccination. Correlation with protection is investigated by challenging vaccinated animals with Beta and Delta SARS-CoV-2 variants of concern. Protection provided by single doses of both vaccines against the omicron variant was also tested. Vaccination resulted in protection as measured by reduction of lung virus titers (sgRNA) and body weight loss (for Ancestral, Beta and Delta VoC).

This study is very timely and addresses an important question: do booster vaccines need to be updated for variants of concern.

Fig. 1: I suggest to choose another acronym for ancestral virus instead of “A”, since this may cause confusion with the Alpha variant.

We thank the reviewer for their comments. We have adjusted all figures to state 'Ancestral' instead of A.

The authors represent viral titers as subgenomic RNA copies in figures 2 and 4. Did authors also quantify replicating virus, for example by TCID₅₀ or plaque assay?

We have now included titration of the swab samples of 3- and 5-days post inoculation, as well as all lung tissue samples.

Why is a different inoculum dose used for the first experiment and the second experiment that includes the omicron variant?

This was due to a relatively low virus stock titer at the time of the experiment. We decided that this was still an appropriate inoculation dose to use, based on within-lab experience published (<https://www.tandfonline.com/doi/full/10.1080/22221751.2020.1858177>) (see figure below).

Rosenke et al. here shows that whether hamsters were inoculated with 10³ or 10⁵ TCID₅₀ does not make a difference for the disease progression.

The high viral loads detected by oropharyngeal swabs compared to the lung virus titers and the effect of vaccination on them should be discussed in the context of transmission studies. Have the authors considered to test the same vaccination regimen on droplet transmission in a transmission study, given their expertise with these models?

We fully agree with the reviewer, and in fact believe this is a very important topic to look into.

We currently have studies ongoing in which hamsters with different immune status (vaccinated IM, vaccinated IN, previously infected) are challenged and subsequently used in transmission chains. This data shows an effect of IM vaccination, an improvement with IN vaccination, and no

to limited transmission with previous infection. See figure below. We are planning to submit this data for publication soon.

The discussion would benefit if authors could elaborate more on how prior vaccination may affect the outcome of booster vaccination. Is this also seen in the clinic or in other preclinical models?

We have added text to the discussion that discusses the potential of original antigenic sin (Line 264-279). We have also included a paragraph that discusses the potential of neutralizing antibodies against vector (Line 280-289).

REVIEWERS' COMMENTS

Reviewer #1 (Remarks to the Author):

The authors have satisfactorily addressed my comments on the first version of the manuscript. I support the publication of the current manuscript.

One minor correction: the word 'obtained' appears twice on line 92.

For future reference, I do have one suggestion to the authors regarding blood collection from hamsters. PMID:19455167 describes blood collection from the cranial vena cava while hamsters are under isoflurane anesthesia. The technique is pain-free and can be repeated multiple times throughout the experiment, unlike retro-orbital puncture.

Reviewer #3 (Remarks to the Author):

All of my concerns have been addressed very well.